# Effect of cellular rearrangement time delays on the rheology of vertex models for confluent tissues

**Gonca Erdemci-Tandogan**[1,2]*, **M. Lisa Manning**[1,3]

**1** Department of Physics, Syracuse University, Syracuse, New York, United States of America, **2** Institute of Biomedical Engineering, University of Toronto, Toronto, Ontario, Canada, **3** BioInspired Institute, Syracuse University, Syracuse, New York, United States of America

* gonca.erdemci@utoronto.ca

**Data Availability Statement:** All relevant data are within the manuscript and its Supporting information files.

## Abstract

Large-scale tissue deformation during biological processes such as morphogenesis requires cellular rearrangements. The simplest rearrangement in confluent cellular monolayers involves neighbor exchanges among four cells, called a T1 transition, in analogy to foams. But unlike foams, cells must execute a sequence of molecular processes, such as endocytosis of adhesion molecules, to complete a T1 transition. Such processes could take a long time compared to other timescales in the tissue. In this work, we incorporate this idea by augmenting vertex models to require a fixed, finite time for T1 transitions, which we call the "T1 delay time". We study how variations in T1 delay time affect tissue mechanics, by quantifying the relaxation time of tissues in the presence of T1 delays and comparing that to the cell-shape based timescale that characterizes fluidity in the absence of any T1 delays. We show that the molecular-scale T1 delay timescale dominates over the cell shape-scale collective response timescale when the T1 delay time is the larger of the two. We extend this analysis to tissues that become anisotropic under convergent extension, finding similar results. Moreover, we find that increasing the T1 delay time increases the percentage of higher-fold coordinated vertices and rosettes, and decreases the overall number of successful T1s, contributing to a more elastic-like—and less fluid-like—tissue response. Our work suggests that molecular mechanisms that act as a brake on T1 transitions could stiffen global tissue mechanics and enhance rosette formation during morphogenesis.

## Author summary

In various morphogenetic events such as embryonic development, tissue repair, or the spread of cancer tumors, cells must move past each other and change neighbors to allow global tissue shape change. In its simplest form, such cell rearrangement events involves neighbor exchanges among four cells, called T1 transitions. During a T1 transition, a sequence of molecular processes must occur over a finite time while cell junctions shrink and new junctions form. In this work, we augment vertex models to require a fixed, finite time for cellular rearrangements, which we call the "T1 delay time". We show that T1

**Funding:** This work was supported by National Institutes of Health under grant R01GM117598 (https://www.nih.gov/) and the Simons Foundation (https://www.simonsfoundation.org) under grants 446222 and 454947 to MLM. The funders had no role in study design, data collection and analysis, decision to publish, or preparation of the manuscript.

**Competing interests:** The authors have declared that no competing interests exist.

delay affects tissue mechanics, stiffening the tissue. We also find that increasing the T1 delay time enhances the percentage of higher-fold coordinated vertices and rosettes, which are seen during many developmental processes such as during the body axis elongation of the fruit fly. Our results highlight the important role of a molecular-scale timescale, T1 delay time, on the global tissue response, and suggest that the organisms might utilize specific molecular processes that act as a brake on cellular rearrangements in order to control the global tissue response.

## Introduction

In processes such as development and wound healing, biological tissues must generate large-scale changes to the global shape of the tissue [1]. In confluent tissues, where there are no gaps or overlaps between cells, such global changes necessarily correspond to either changes in individual cell shape or cell rearrangements [2], and large-scale deformation almost always requires a large number of rearrangements.

For epithelial monolayers, the geometry of most such rearrangements is quite simple: viewing the apical side of the layer, four cells come together at a single four-fold vertex, which subsequently resolves into two three-fold vertices where cells have exchanged neighbors. This process is called a T1 transition, adopted from the literature on foams [3]. In some tissues, it is also common to observe higher-fold vertices called rosettes [4, 5].

Historically, there have been different perspectives on how to understand and quantify such rearrangements. At the *molecular scale*, a concerted sequence of processes must occur to allow such a change, including localization of non-muscle myosin and actin to shorten interfaces [6–10], unbinding of adhesion molecules and trafficking away from the membrane via endocytosis [11], exocytosis of adhesion molecules to newly formed interfaces and new homotypic binding, and reorganization of the cytoskeleton to stabilize the new edges. Moreover, molecules such as tricellulin [12, 13] are known to localize at tricellular junctions and must be reorganized [14, 15]. In addition to all this, there is recent evidence that some cell types possess mechanosensitive machinery that will only trigger this molecular rearrangement cascade if tension on the interface is sufficiently large [16].

A complementary perspective has focused on the *collective behavior* of cells in a tissue. Specifically, a combination of theoretical modeling [17, 18] and experimental data [19–21] has suggested that the collective mechanics of a tissue has a huge impact on the rate of cell rearrangements, and that the collective mechanics are dominated by a simple observable, the cell shape.

Shapes of cells in a confluent tissue are generated by a balance between contractility generated by the cytoskeleton and adhesion generated by molecules such as cadherins, as well as active force generation by cells [21, 22]. This suggests that cell rearrangement rates are governed by cell-scale features, such as overall expression levels of adhesion and cytoskeletal machinery. Moreover, it is possible to identify a "collective response" timescale $\tau_{\alpha 0}$ [23] that describes the typical timescale over which cells change neighbors. In both isotropic and anisotropic tissues, this timescale depends on cell shape and alignment [19, 21, 22].

Given the strong experimental support for cell-scale and molecular-scale perspectives, we hypothesize that both kinds of mechanisms must be working in concert to drive cell rearrangement rates in confluent tissues.

While vertex models have been successful in predicting rearrangement rates in many cases, standard versions of vertex models are missing the fact that specific molecular cascades and

triggers are needed to allow rearrangements. Specifically, standard vertex model formulations require that cell neighbor exchanges proceed instantaneously after the creation of a higher-fold coordinated vertex. In systems where molecular mechanisms delay cell neighbor exchanges after the creation of higher-fold vertices, vertex models will make incorrect predictions for the global tissue response.

An extreme example is the amnioserosa tissue that is required for proper germband extension in *Drosophila* [24]; although the cell shapes become extremely elongated in the tissue [25], and standard vertex models would predict high numbers of cell rearrangements in response, experiments demonstrate that cells do not change their neighbor relationships at all. This is important for amnioserosa function: an elastic response generated when cells maintain neighbor relationships allows the amnioserosa to pull strongly on the germband tissue and elongate it [24, 26, 27].

While experimental work is ongoing to understand the precise molecular mechanisms that prevent cell rearrangements in the amnioserosa, there have been recent attempts to augment vertex models by incorporating various molecular mechanisms that affect how cells exchange neighbors. One model already highlighted above studies how a stress or strain threshold for T1 transitions affects cell shapes and rates of cell rearrangement [16]. Other models incorporate strongly fluctuating line tensions [28, 29] that can trap edges so they cannot execute T1 transitions.

In this manuscript, we instead augment vertex models with a model parameter we term the "*T1 delay time*", which is intended to incorporate a broad range of molecular mechanisms that act as a brake on T1 transitions. For every situation where the cell-scale dynamics generate a four-fold or higher coordinated vertex, the vertex is prevented from undergoing a T1 transition for a duration we call the T1 delay time. A similar mechanism has also been studied in independent concurrent work by Das *et al* [30], which focuses on how controlled T1 timescales generate intermittency and streaming states in glassy isotropic tissues. Here, we study how such a mechanism alters the global response of a tissue, both in isotropic tissues and in tissues where there is a global anisotropic change to tissues shape, such as during convergent extension in development.

We find that the "molecular-scale" T1 delay timescale dominates over the "cell-scale" collective response timescale when the T1 delay timescale is the larger of the two, slowing down tissue dynamics and solidifying the tissue. In addition, we find that increasing T1 delay time enhances the rosette formation in anisotropic systems. This suggests that organisms might utilize specific molecular processes that act as a brake on the resolution of four-fold and higher coordinated vertices in order to control the global tissue response in processes such as wound healing and convergent extension.

## Model

### Vertex model with T1 rearrangement time

We introduce a new model parameter, T1 delay time, both in isotropic and anisotropic vertex models. A vertex model defines an epithelial tissue as confluent tiling of $N$ cells with an energy functional for the preferred geometries of the cells [17, 22, 31]

$$E = \sum_i^N K_A (A_i - A_{0i})^2 + K_P (P_i - P_{0i})^2. \qquad (1)$$

In this definition, both the area and perimeter of a cell act like an effective spring. Here, $A_i$ and $A_{0i}$ are the actual and preferred areas of cell $i$ while $P_i$ and $P_{0i}$ are the actual and preferred

perimeters. Using open-source cellGPU software [32], we simulate over-damped Brownian dynamics, where the positions of the cell vertices are updated at each time step according to

$$\Delta r_i^\alpha = \mu F_i^\alpha \Delta t + \eta_i^\alpha \qquad (2)$$

using a simple forward Euler method. Here $F_i^\alpha = -\nabla_i E$ is the force on vertex $i$ in $\alpha$ direction, $\mu$ the inverse friction, $\Delta t$ the integration time step and $\eta_i^\alpha$ is a normally distributed random force with zero mean and $\langle \eta_i^\alpha(t)\eta_j^\beta(t')\rangle = 2\mu T \Delta t \delta_{ij}\delta_{\alpha\beta}$. The temperature T sets a thermal noise on vertices of each cell. The integration time step is set to $\Delta t = 0.01\tau$ where $\tau$ is the natural time unit of the simulations: $\tau = 1/(\mu K_A A_0)$.

As is standard in the literature, for instantaneous cell-neighbor exchanges, a T1 transition proceeds whenever the distance between two vertices is less than a threshold value, $l_c = 0.04$ in natural simulation units. We have checked that our results are not sensitive to the precise value of this cutoff. We set $K_A = 1$, $K_P = 1$, $\mu = 1$ and all cells to be identical so that $A_{0i} = A_0$ and $P_{0i} = P_0$. We nondimensionalize the length by the natural unit length of the simulations $l = \sqrt{A_0}$. This yields a target shape index or a preferred perimeter/area ratio $p_0 = P_0/\sqrt{A_0}$.

The T1 delay time, $t_{T1}$, is a finite T1 rearrangement time that acts as a brake on T1 transitions, which could arise from a broad range of molecular mechanisms as discussed in the introduction. While there are many ways of implementing such a feature in our model, including adding an explicit additional energy barrier to that defined in Eq 1, for simplicity we add the delay directly to the system dynamics. Specifically, when the cell-scale dynamics generate a four-fold or many-fold vertex (cellular junctions that satisfy $l < l_c$ criteria), the vertex is prevented from undergoing a T1 transition for the duration of the T1 delay time, $t_{T1}$ (Fig 1A and 1B). While the edge waits for a $t_{T1}$ time, the configuration is not on hold—the system evolves according to Eq 2 and edges can still lengthen and shorten. In particular, every edge of a cell is associated with two timers (one for each connected vertex) to keep track of the time delays. When the timer reaches $t_{T1}$, if an edge $l$ is still less than a critical length $l_c$, the associated cell undergoes the T1 process.

## Anisotropic vertex model

We introduce anisotropy in our model using two different sets of simulation methods, aniosotropic line tensions and shear, similar to the protocols described in [21]. For the first set of simulations, we introduce an additional line tension to nearly vertically-oriented edges. This type of perturbation was first developed to model dynamic anisotropic myosin distribution due to planar cell polarity pathways in the germband extension in *Drosophila* [9, 33]. This is implemented via a vertex model energy functional:

$$E = \sum_i^N K_A(A_i - A_{0i})^2 + K_P(P_i - P_{0i})^2 + \sum_{<j,k>}\gamma_{<j,k>}l_{<j,k>}. \qquad (3)$$

Here, the first sum is same as Eq 1, while the second sum introduces an additional anisotropic line tension, summed over all edges connecting vertices $j$ and $k$. $l_{<j,k>}$ is the length of the edge between vertices $j$ and $k$, and $\gamma_{<j,k>}$ is a line tension specified as

$$\gamma_{<j,k>} = \gamma_0 \cos\left[2(\theta_{<j,k>} - \phi)\right], \qquad (4)$$

where $\gamma_0$ is the amplitude, $\theta_{<j,k>}$ is the edge angle and $\phi$ is the angle of anisotropy. The line tension will be maximum for the edges parallel to the lines with angle $\phi$ and will be minimum for the edges that are perpendicular to $\phi$. In the following, we fix $\phi = \pi/2$.

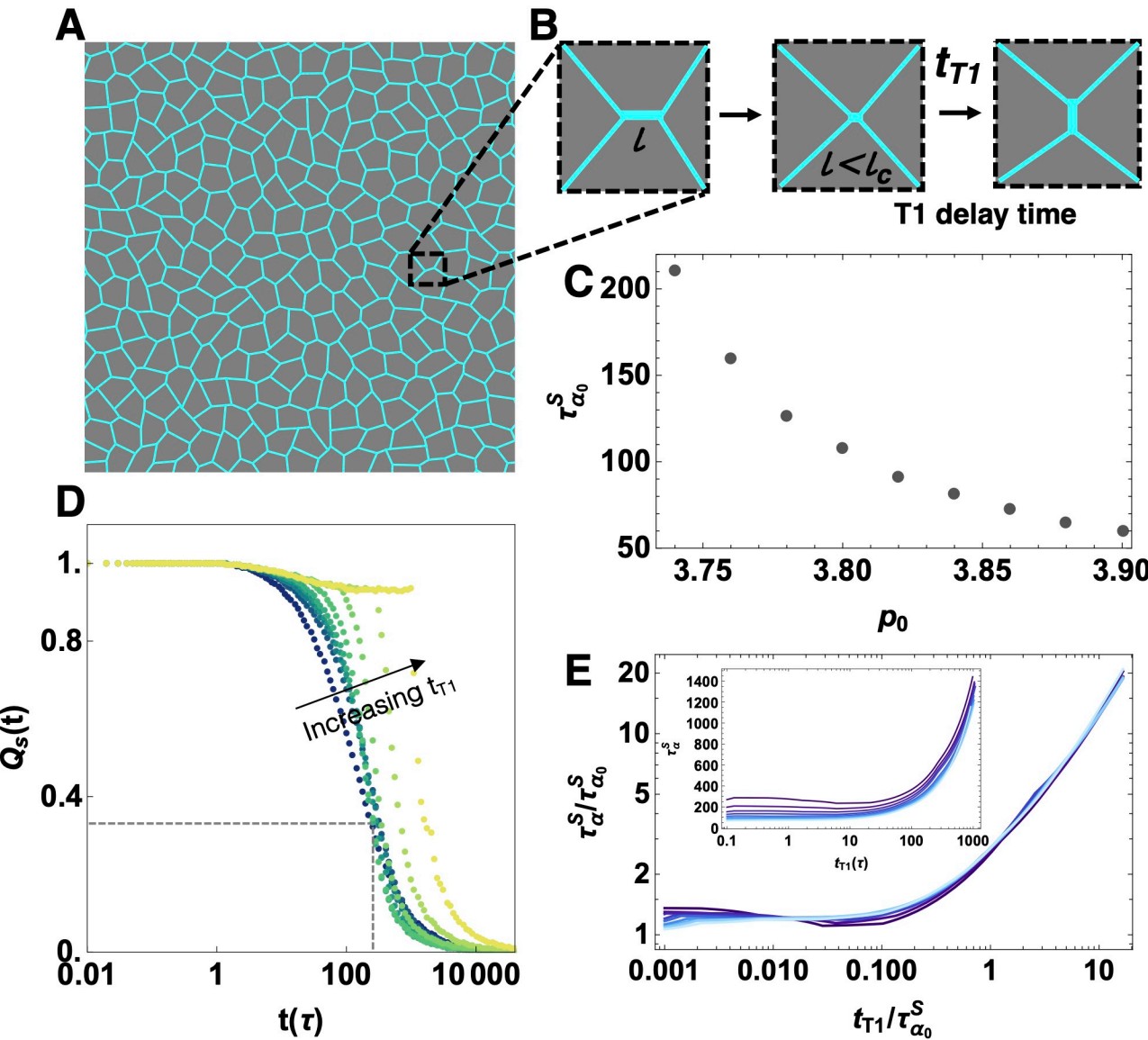

**Fig 1. Mechanical response of the isotropic vertex model at finite temperature as a function of T1 delay time.** A) Example cell configuration in an isotropic vertex model at a finite temperature $T = 0.02$ and fixed system size $N = 256$. B) Schematic of a cellular rearrangement. An edge length of $l$ shrinks to a length less than a critical length $l_c$, forming a four-fold vertex. The edge is prevented from undergoing the T1 transition for a $t_{T1}$ delay time as described in the main text. C) The characteristic relaxation time in the absence of T1 delays, defined by the self-overlap function, for various $p_0$ values. The tissue becomes more viscous as $p_0$ decreases at fixed temperature. D) Self-overlap function for T1 rearrangement delay time of $t_{T1} = 0, 0.13, 0.46, 1.67, 5.99, 21.5, 77.4, 278.2$ and $1000\tau$ (darker green to yellow) for $p_0 = 3.74$. The dotted lines indicate where $Q_s(\tau_{\alpha 0}^S) = 1/e$ in the absence of a T1 delay. E) Log-log plot showing collapse of the characteristic relaxation time $\tau_\alpha^S$ as a function of T1 delay time normalized by the collective response timescale $\tau_{\alpha 0}^S$ without a T1 delay. Colors correspond to different values of $p_0 = 3.74, 3.76, 3.78...3.9$ (darker to light blue), for fixed $T = 0.02$, and $N = 256$. The inset shows the characteristic relaxation time $\tau_\alpha^S$ as a function of T1 delay time without any normalization, for the same values of $p_0$.

For the second set of anisotropic simulations, we apply an external pure shear on the simulation box which is initially a square domain with $L_x = L_y = L$. We shear the box such that $L_x = e^\epsilon L$ and $L_y = e^{-\epsilon}L$. Here, $\epsilon$ is the shear strain. For the initial square domain, $\epsilon = 0.0$, then we increase it by $5 \times 10^{-6}$ increments at every simulation step. After each shear, we minimize the system by updating the vertex positions but keeping the box dimensions fixed.

## Results

### Relaxation time of the tissue

To quantify the global mechanical properties of the tissue, we characterize a relaxation time ($\tau_\alpha^s$) as a function of T1 delay time $t_{T1}$ and target shape index $p_0$ using the decay of a self-over-lap function. The self-overlap function is a standard correlation function used to quantify glassy dynamics in molecular and colloidal materials [34]. It represents the fraction of particles (vertices) that have been displaced by more than a characteristic distance $a$ in time t,

$$Q_s(t) = \frac{1}{N}\sum_{i=1}^{N} w(|\mathbf{r}_i(t) - \mathbf{r}_i(0)|) \tag{5}$$

where $\mathbf{r}_i$ is the position of vertex $i$ and the function $w$, $w(r \leq a) = 1$ and $w(r > a) = 0$. The characteristic relaxation time of the system, $\tau_\alpha^s$, is the time which most of the vertices are displaced more than a characteristic distance: $Q_s(\tau_\alpha^s) = 1/e$.

We run simulations across a three-dimensional parameter space: T1 delay time $t_{T1}$, target shape index parameter $p_0 = P_0/\sqrt{A_0}$, and temperature $T$, with 100 independent simulations initialized from different configurations for each point in parameter space. All simulations are thermalized at their target temperature for $10^4$ $\tau$ before recording the data. We then run simulations for additional $3 \times 10^5$ $\tau$ to ensure the system reaches a steady state.

We first use the self-overlap function to compute the characteristic relaxation time of the tissue, $\tau_{\alpha 0}^s$, in the absence of any T1 delays. This is simply the time at which the self-overlap function decays to $1/e$ of its original value for simulations where the T1 rearrangement is instantaneous, $t_{T1} = 0$. Therefore, $\tau_{\alpha 0}^s$ is the typical collective response timescale that depends on cell shape and alignment in vertex models. Fig 1C shows this timescale as a function of the target shape parameter $p_0$ with $T = 0.02$. As expected, $\tau_{\alpha 0}^s$ decreases monotonically as $p_0$ increases, demonstrating that lower values of $p_0$ are associated with glassy behavior and increasing relaxation times.

Next, we study the behavior of the self-overlap function as a function of the T1 delay time. Fig 1D shows that the self-overlap function changes a function of the T1 delay time, with $t_{T1} = 0, 0.13, 0.46, 1.67, 5.99, 21.5, 77.4, 278.2$ and $1000$ $\tau$ (dark green to yellow) for fixed $p_0 = 3.74$ and $T = 0.02$. From this data and additional simulations at other values of $p_0$, we extract the characteristic characteristic relaxation time in the presence of T1 delays, $\tau_\alpha^s$. The inset to Fig 1E shows the behavior of $\tau_\alpha^s$ as a function of $t_{T1}$ for different values of $p_0 = 3.74, 3.76, 3.78\ldots3.9$ (darker to light blue), $T = 0.02$ and $N = 256$.

As $\tau_{\alpha 0}^s$ represents the inherent relaxation timescale of the tissue controlled by $p_0$ in the absence of T1 delays, we attempt to collapse this data by rescaling both the T1 delay timescale $t_{T1}$ and the observed relaxation timescale $\tau_\alpha^s$ by the inherent timescale for each value of $p_0$. The data collapses, showing that the mechanical properties of the tissue remain unchanged for any $t_{T1}$ delay time below $\sim 10\%$ $\tau_{\alpha 0}^s$. For $t_{T1} \gtrsim 10\%$ $\tau_{\alpha 0}^s$, the relaxation time increases significantly, with a slope approximately equal to unity, indicating $\tau_\alpha^s \approx t_{T1}$ when $t_{T1} > 10\%$ $\tau_{\alpha 0}^s$. The best linear fit (S1 File) to this region has a coefficient $m = 1.13 \sim 1$, indicating that the relaxation time is just the T1 delay time: $\tau_\alpha = t_{T1}$. This suggests that when the "molecular-scale" T1 delay timescale is larger than the cell-scale collective timescale $\tau_{\alpha 0}^s$, it dominates the response and solidifies the tissue. Moreover, the data collapse suggests that the cellular rearrangement time-scale $\tau_{\alpha 0}^s$ is a good proxy for the mechanical response of the tissue over a wide range of model parameters. We note that these results are not dependent on the system size, as shown in S1 File.

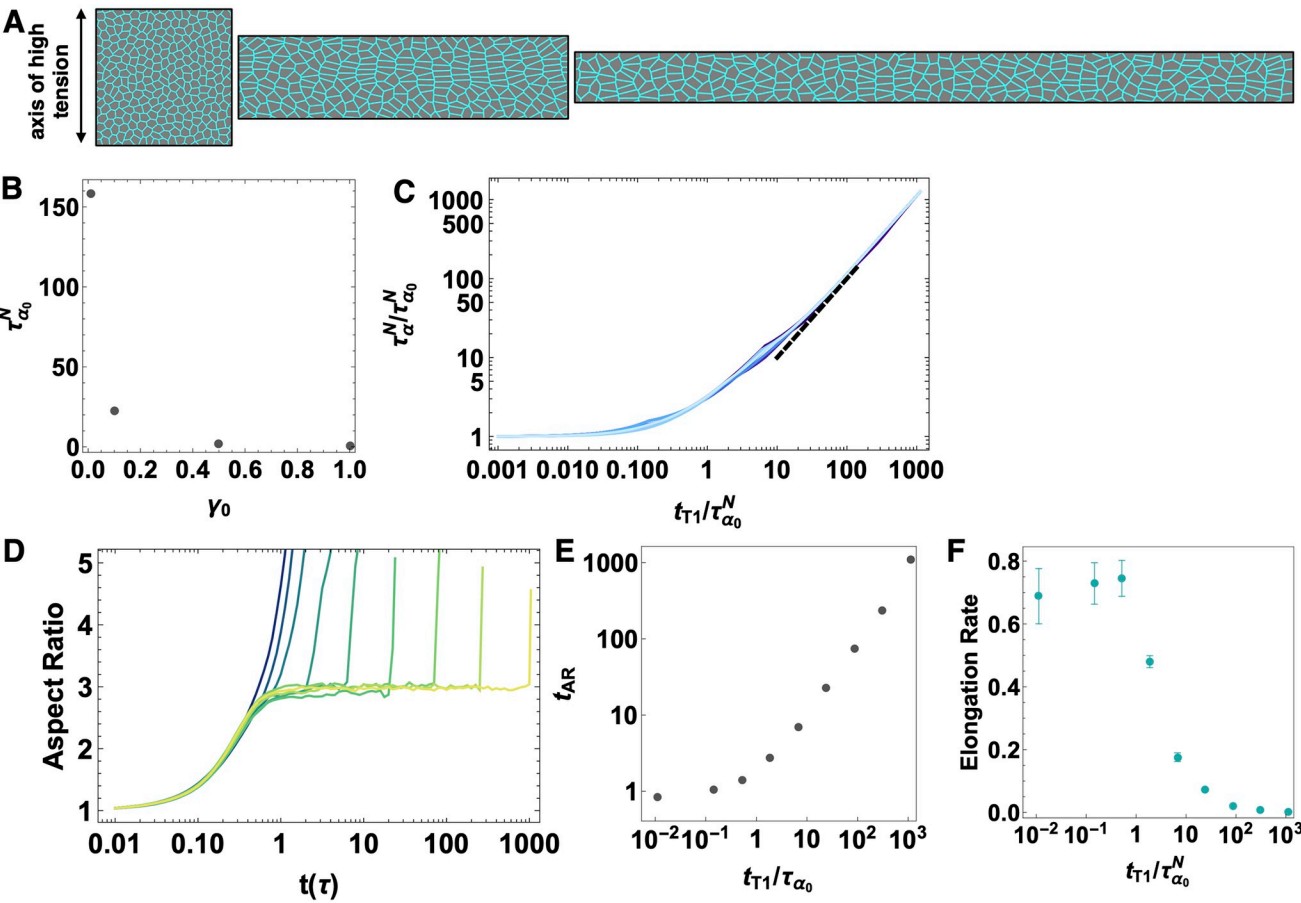

**Fig 2. Anisotropic vertex model with T1 delay time.** A) Simulations of an anisotropic tissue. An anisotropic line tension on vertical edges is introduced to obtain global anisotropic changes to tissue shape. B) The collective response time scale for various $\gamma_0$ values, the anisotropic line tension amplitude. C) Data collapse for $p_0 = 3.74, 3.76, 3.78...3.9$ (darker to lighter blue), $T = 0.02$, $N = 256$ and $\gamma_0 = 1.0$. The characteristic relaxation time, $\tau_\alpha^N$ as a function of T1 rearrangement delay time normalized by the collective response timescale $\tau_{\alpha 0}^N(t_{T1} = 0)$. The dotted line is a slope of 1. D) The aspect ratio of the simulation box over time for T1 delay time of $t_{T1} = 0, 0.13, 0.46, 1.67, 5.99, 21.5, 77.4, 278.2$ and $1000\ \tau$ (dark green to yellow), $p_0 = 3.74$, $T = 0.02$, $N = 256$ and $\gamma_0 = 1.0$. E) The time ($t_{AR}$) at which the system first goes above the plateau value as a function of $t_{T1}$ for each aspect ratio curve in (D). F) The rate of elongation obtained from the aspect ratio curves in (D) as a function of $t_{T1}$ delay time. (D), (E) and (F) are from 10 independent simulation runs and the rate values are average ± one standard error.

## Convergent extension rate is disrupted by T1 rearrangement time

As many biological processes require large-scale, anisotropic changes in tissue shape, we next focus on the role of T1 delays in models that are anisotropic.

We first study the dynamics of a vertex model with anisotropic line tensions, described by Eqs 3 and 4, where we initialize the simulations on a square domain. Fig 2A shows snapshots of the evolution of an anisotropic tissue in our simulations.

For the anisotropic tissue, we first note that the standard overlap function $Q_s$ defined in Eq 5 is not a good metric for the rheology of a tissue with global shape changes or tissue flow. This is because cells may stop overlapping their initial positions due to the macroscopic flow instead of due to local neighbor exchanges that are important for rheology. Therefore, we use a different neighbors-overlap function $Q_n$ [35] to capture the rheology, which represents the fraction of cells that have lost two or more neighbors in time $t$ (see S1 File for further details). In isotropic systems, $Q_n$ and $Q_s$ are very similar, but $Q_n$ is much better at identifying rheological

changes in anisotropic systems. Formally, it is defined as

$$Q_n(t) = \frac{1}{N}\sum_{i=1}^{N} w \qquad (6)$$

where $w = 0$ if a cell has lost two or more neighbors and $w = 1$ otherwise. Then the characteristic relaxation time of the system measured by the nearest neighbors overlap function, $\tau_\alpha^N$, is the time when $Q_n(\tau_\alpha^N) = 1/e$.

As in the isotropic case, we run simulations across a range of $t_{T1}$, $p_0$, and temperature $T$, with 100 independent simulations at each point in parameter space, and where all simulations are thermalized at their target temperature for $10^4\ \tau$. We then run simulations for additional simulation time until the box reaches to a height of about four cells to avoid numerical instabilities due to periodic boundary conditions.

First, we find that the characteristic relaxation $\tau_{\alpha 0}^N$ is controlled not only by $p_0$ but also by the magnitude of the applied anisotropic line tension in Eq 4, $\gamma_0$. This data is shown for fixed $p_0 = 3.74$ in Fig 2B.

Fig 2C illustrates the characteristic relaxation time, $\tau_\alpha^N$, for the same values of $p_0$ and $T$ as shown in Fig 1E, but with a fixed anisotropic line tension amplitude of $\gamma_0 = 1.0$. Both axes are normalized by the collective response timescale $\tau_{\alpha 0}^N$ which corresponds to the case where the T1 rearrangement is instantaneous, $t_{T1} = 0$. The relation between the molecular-scale T1 delay timescale and collective response timescale is similar to that of the isotropic tissue, where the molecular-scale T1 delay timescale dominates over the cell based collective response timescale when the $t_{T1}$ delay time is larger than the collective response timescale. S1 File shows that this result is independent of the magnitude of the line tension in the anisotropic model. We also note that the characteristic relaxation time behavior is the same at zero temperature (S1 File).

Next we analyze the rate of convergent extension for a fixed $p_0 = 3.74$ and $\gamma_0 = 1.0$ value, in order to study the role of T1 delays on tissue-scale deformations. Fig 2D is a plot of the aspect ratio of the simulation box over time for different values of $t_{T1}$. We see that for $t_{T1} \lesssim \tau_{\alpha 0}^N$, there is a smooth elongation process until the simulation ends, consistent with a fluid-like response (or like the behavior of a yield-stress solid above the yield stress.) In this regime, increasing $t_{T1}$ increases the rate of elongation slightly. In contrast, for $t_{T1} \gtrsim \tau_{\alpha 0}^N$, the system first plateaus at a specific aspect ratio (which is about three for the parameter values shown here), and only begins to elongate beyond that value for timescales greater than $t_{T1}$. The plateau value occurs in the absence of any rearrangements, so it is entirely due to changes in individual cell aspect ratios. It is therefore governed by a balance between $\gamma_0$ and $k_P P_0$, and can be predicted analytically, as described in the S1 File. These features are not strongly dependent on system size, as shown in S1 File.

The change in behavior at $t_{T1} \sim \tau_{\alpha 0}^N$ is highlighted in the inset of Fig 2E, where we plot the time ($t_{AR}$) at which the system first goes above the plateau value as a function of $t_{T1}$. Similar features can be seen in a plot of the elongation rate as a function of T1 delay time, shown in Fig 2F. We calculated the rate of elongation (Fig 2F) as the growth constant of an exponential fit to the aspect ratio over time (Fig 2D) for each T1 delay time value.

To see if these observations are specific to systems where the anisotropy is generated by internal line tensions, or instead a generic feature of anisotropic systems, we study a vertex model in the presence of an externally applied pure shear strain (Fig 3A). Fig 3 shows the collective response of the tissue to pure shear. Fig 3B illustrates the inherent relaxation timescale extracted from $Q_n$ as a function of $p_0$, and the fact that is very similar to that in Fig 1C suggests both that $Q_n$ and $Q_s$ are providing similar information and that the tissue rheology is robust across different perturbations (fluctuations vs. shear). Fig 3C is similar to both Fig 2C and

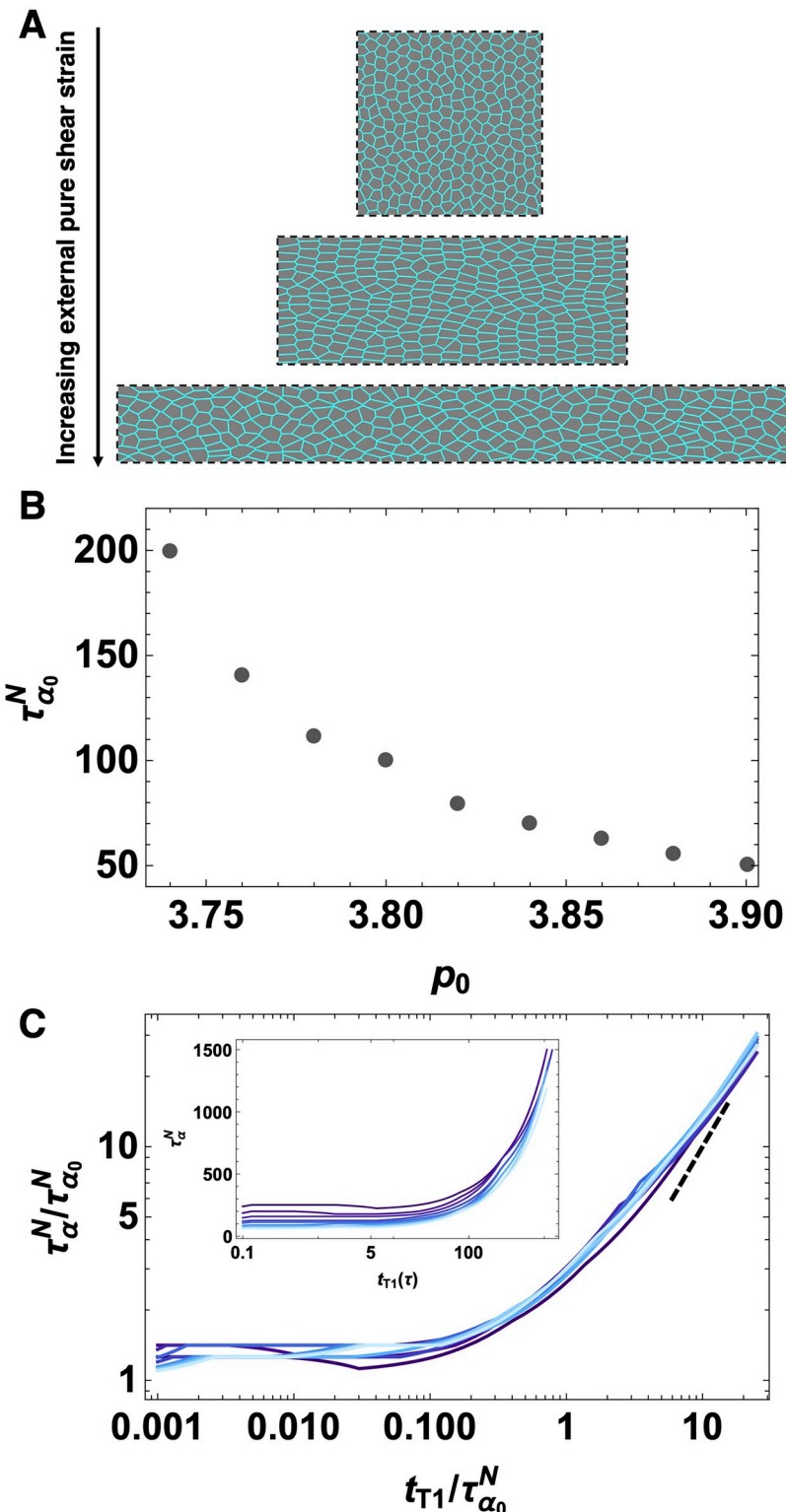

**Fig 3. Anisotropic vertex model with external pure shear and T1 delay time.** A) We apply an external pure shear on the simulation box which is initially a squared domain of $L_x = L_y = L$. We shear the box such that $L_x = e^{\epsilon} L$ and $L_y = e^{-\epsilon} L$. Snapshots are from simulations with $p_0 = 3.74$, $T = 0.02$, $N = 256$ and $t_{T1} = 1000\tau$. B) The collective response time scale $\tau_{\alpha_0}^N$ ($t_{T1} = 0$) for various $p_0$ values. C) Data collapse for $p_0 = 3.74, 376, 3.78 \dots 3.9$ (darker to lighter blue), $T = 0.02$ and $N = 256$. The characteristic relaxation time, $\tau_{\alpha}^N$ as a function of T1 rearrangement delay time normalized by the collective response timescale $\tau_{\alpha_0}^N(t_{T1} = 0)$. The dotted line is a slope of 1. Inset shows $\tau_{\alpha}^N$ the characteristic relaxation time as a function of T1 delay time before normalization for p0 = 3.74, 3.76, 3.78 . . .3.9 (darker to light blue).

Fig 1E, confirming that our observation that T1 delays do not affect the tissue mechanics until $t_{T1} \sim 10\% \ \tau_{\alpha_0}$, and beyond that value the tissue mechanics is dominated by that timescale, is robust across all perturbations studied.

## T1 rearrangement time delays and tissue anisotropies contribute to rosette formation

While T1 transitions are the simplest type of rearrangements in epithelial monolayers, it is common for vertices that connect more than 4 cell edges to appear during developmental processes. These are termed "rosettes" and they appear often in tissue morphogenesis [4, 5] and collective cell migration [36].

Although our model only allows three-fold coordinated vertices, previous work by some of us has shown that vertices connected by short edges can be considered as a proxy for higher-order coordinated vertices [29]. In that work, a cutoff of $0.04\sqrt{A_0}$ was used to threshold very short edges as a proxy for multi-fold coordination. Something similar is also explicitly the case in experiments, where due to microscope resolution it is not possible to distinguish between very short edges and multi-fold coordinated vertices [21]. In that work, a cutoff of $0.11\sqrt{A_0}$ was imposed by microscope resolution, and also adopted in analysis of vertex models. Moreover, many-fold vertices are shown to be stable at heterotypic interfaces [37].

In our simulations of tissues with anisotropic line tensions, either the additional anisotropic tension on interfaces, or the edges that are prevented from undergoing T1 transitions, or both, could generate an increase in the number of observed rosettes. Therefore, to study the role of T1 time delays on higher-order vertex formation, we analyze the number of very short edges per cell over time in our simulations. For figures in the main text we adopt the larger cutoff of $0.11\sqrt{A_0}$ used in [21], but in the supporting information (S1 File) we show that the results remain qualitatively similar for the smaller cutoff of $0.04\sqrt{A_0}$ used in [29], although of course the overall number of very short edges is smaller with the lower threshold.

Fig 4A–4C highlights snapshots of typical cellular structures at different timepoints in an anisotropic simulation with intermediate T1 delay time. Initially (panel A), cells are isotropic, and after about one natural time unit (panel B) vertically oriented edges under anisotropic tension have shrunk to near zero length, resulting in a significant number of 4-fold coordinated vertices and elongated rectangular shapes with an aspect ratio set by a balance of anisotropic ($\gamma_0$) and isotropic ($\kappa_P P_0$) tensions, as discussed in S1 File. At timescales larger than the T1 delay time (panel C), T1 transitions allow some short edges to resolve and relax the structure.

Fig 4D shows the number of short edges (SE) per cell ($\xi = 2 * N_{SE}^{avg}/N_{cell}$, where the factor of two reflects that edges are shared by two cells) over time for T1 delay of $t_{T1} = 0, 0.13, 0.46, 1.67, 5.99, 21.5, 77.4, 278.2$ and $1000 \ \tau$ for an anisotropic tissue generated using anisotropic internal tensions. For all values of the T1 delay time, there is an initial sharp rise to a maximum value, followed by a decrease. This decrease suggests that, although there is a significant population of higher-fold vertices that remain unresolved when the T1 delay is large, some many-fold vertices resolve on longer timescales. This aligns well with experimental observations of germ-band extension in *Drosophila*, where cell junctions with dorsal-ventral orientation collapse to form higher order rosette structures and the rosettes are resolved by the extension of new junctions in anterior-posterior orientation [38].

In addition, Fig 4E shows that as the T1 delay time increases, the maximum number of short edges increases. Again, we see that there is a change in behavior around $t_{T1} \sim \tau_{\alpha 0}^N$. For $t_{T1} \lesssim \tau_{\alpha 0}^N$, it is a monotonically increasing function, while for $t_{T1} \gtrsim \tau_{\alpha 0}^N$, it plateaus at the same large value of about 1 edge per cell. This is consistent with our previous discussion of the

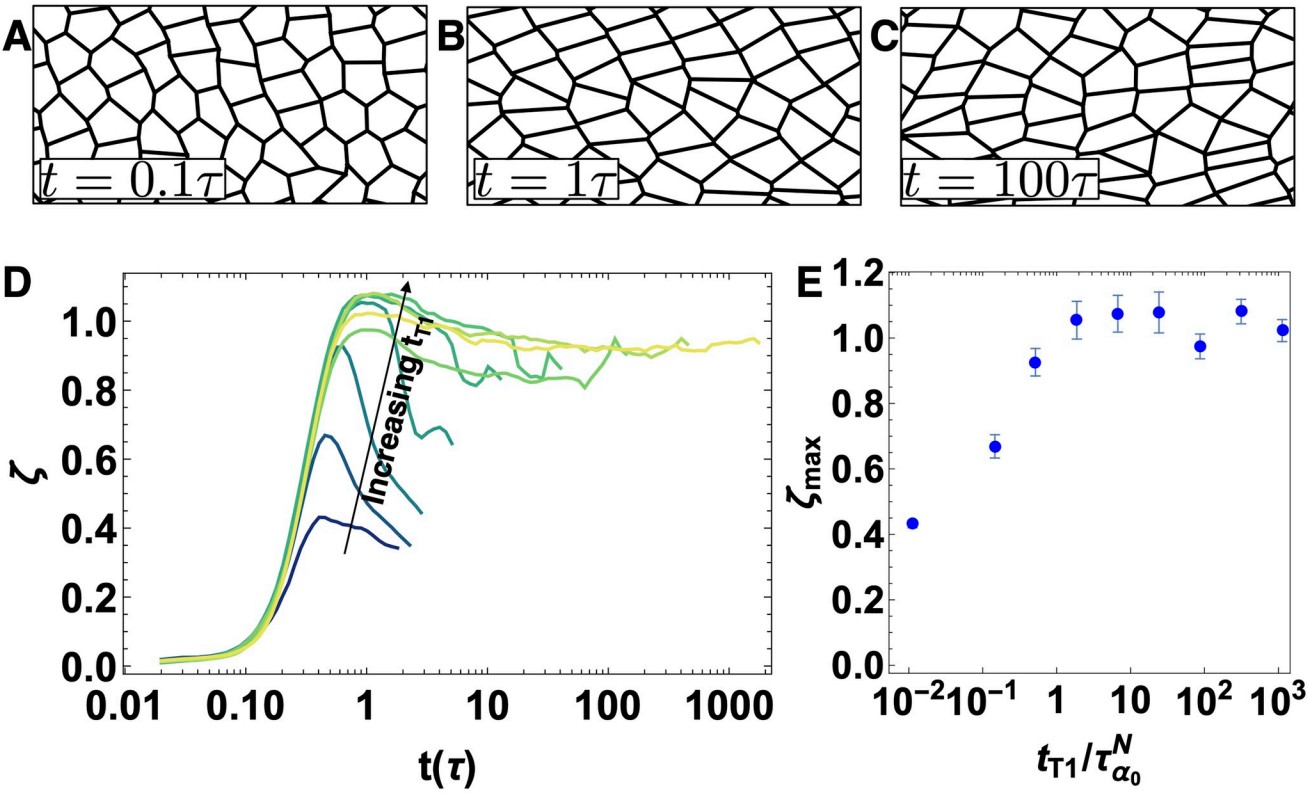

**Fig 4. Counting very short edges as a proxy for many-fold vertices.** Snapshots of configurations at (A) $t = 0.1$, (B) $t = 1$, and (C) $t = 100$ $\tau$ for $t_{T1} = 77.4$ $\tau$ for a tissue with $p_0 = 3.74$, $T = 0.02$, $N = 256$ and $\gamma_0 = 1.0$. D) Number of very short edges per cell $\xi$ for an anisotropic tissue as a function of time. Shaded lines represent different T1 delay times $t_{T1} = 0, 0.13, 0.46, 1.67, 5.99, 21.5, 77.4, 278.2$ and $1000$ $\tau$ (dark green to yellow), for a tissue with $\tau_{\alpha_0}^N = 0.89\tau - p_0 = 3.74$, $T = 0.02$, $N = 256$ and $\gamma_0 = 1.0$. E) Ensemble-averaged maximum value of $\xi$ over a simulation timecourse ($\xi_{max}$) vs. the T1 delay time $t_{T1}$ normalized by $\tau_{\alpha_0}^N$. The average is taken over 10 independent simulations, and error bars correspond to one standard error.

mechanisms driving elongation, where we noted that for large $t_{T1}$ there were no cellular rearrangements until $t = t_{T1}$, and so prior to that timepoint the cells individually deform until they form a nearly rectangular lattice. A perfect rectangular lattice would have $\xi_{max} = 2$, so that two edges of the hexagon have shrunk to zero length, whereas in our disordered systems we find approximately one short edge per cell. Nevertheless, the maximum in the number of short edges is associated with these maximally deformed cell shapes.

We also find that in isotropic tissues, increasing the T1 delay times increases the number of many-fold vertices (S1 File), although the numbers per cell are much smaller than that in the anisotropic tissue, as expected.

All together, our results suggest that anisotropic line tension can collapse cell-cell junctions, resulting higher number of many-fold structures. The number of such structures increases as the T1 rearrangement time delay increases. Therefore, cellular rearrangement time could be a mechanism to regulate multicellular rosette structures during morphogenesis.

## Number of T1 rearrangements

To study the impact of T1 delays on number of T1 rearrangements, we calculate the number of *successful* T1 transitions per cell ($\eta = N_{T1(irr)}^{avg}/N_{cell}$) over time for a T1 delay of $t_{T1} = 0, 0.13,$ $0.46, 1.67, 5.99, 21.5, 77.4, 278.2$ and $1000$ $\tau$ (Fig 5). In particular, in our model if an edge length $l$ is less than the critical length $l_c$ and the associated T1 delay timer reaches to zero, the

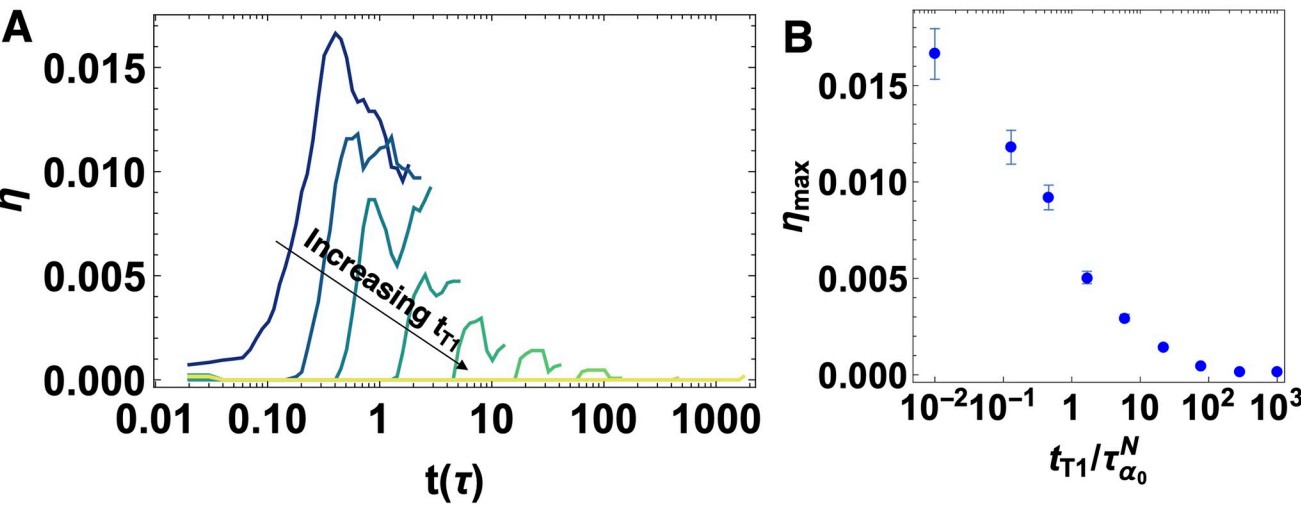

**Fig 5. Number of successful T1 transitions.** A) Number of successful (irreversible) T1 transitions per cell $\eta$ as a function of simulation time $t$ for an anisotropic tissue with T1 delay time of $t_{T1} = 0$, 0.13, 0.46, 1.67, 5.99, 21.5, 77.4, 278.2 and 1000 $\tau$ (dark green to yellow), with $\tau_{\alpha 0}^N = 0.89\tau - p_0 = 3.74$, $T = 0.02$, $\gamma_0 = 1.0$ and $N = 256$, averaged over 10 independent realizations. B) Number of successful (irreversible) T1 transitions at the maximum averaged over 10 realizations. Error bars represent one standard error.

edge orientation is flipped. However, the same edge can flip back and forth to its original orientation easily until its final steady state condition is obtained. Hence, we define successful T1 transitions as the arrangements that the cells rearrange and stay in their new configurations. In other words, the arrangements are irreversible (see S1 File for details).

The data for the the number of successful (irreversible) T1 transitions share some similarities with the analysis of short edges in Fig 4. Specifically Fig 5A shows that the number of successful T1s grows towards a maximum and then decays, which is consistent with a picture that cells first become deformed and then execute T1 transitions at longer timescales to facilitate large-scale tissue deformation. Fig 5B shows that this maximum decreases rapidly with increasing T1 delays, highlighting that the tissue response is much more elastic-like in the limit of large T1 delays. Again, we see a crossover in behavior around $t_{T1} \sim \tau_{\alpha 0}^N$. Similarly, we study the effect of T1 delays on the number of T1 transitions for an isotropic set of simulations. The number of successful T1 transitions decreases as T1 delays increase (S1 File), although the effect is much weaker in isotropic tissues as there are no large-scale deformations driving T1 transitions in that case.

## Viscoelastic behavior of cellular junctions

Although the previous sections focus on global tissue response, we wanted to briefly investigate the impact of T1 delays on localized cellular junction dynamics. We apply a contractile tension on a cellular junction in a simulation with anisotropic internal tension. Specifically, we start from a configuration in the final steady state for a given $\gamma_0$ and $p_0$, and we choose $p_0 = 4.0$ to be in the fluid-like regime in the absence of T1 delays. We apply a large (Fig 6A) or small (Fig 6B) contractile stress on a edge, while ensuring that the stress is not large enough to generate T1 transitions. We then remove the stress from the edge after a fixed time period and record the global tissue response. After the stress is removed, the edge recovers a small amount but remains permanently deformed (Fig 6A and 6B blue curves). We repeat the same procedure for a simulation with a large T1 time delay $t_{T1} = 278.2\tau$; in this case the tissue exhibits viscoelastic features and recoils and recovers back to %70–%90 of its initial length in high (Fig 6A

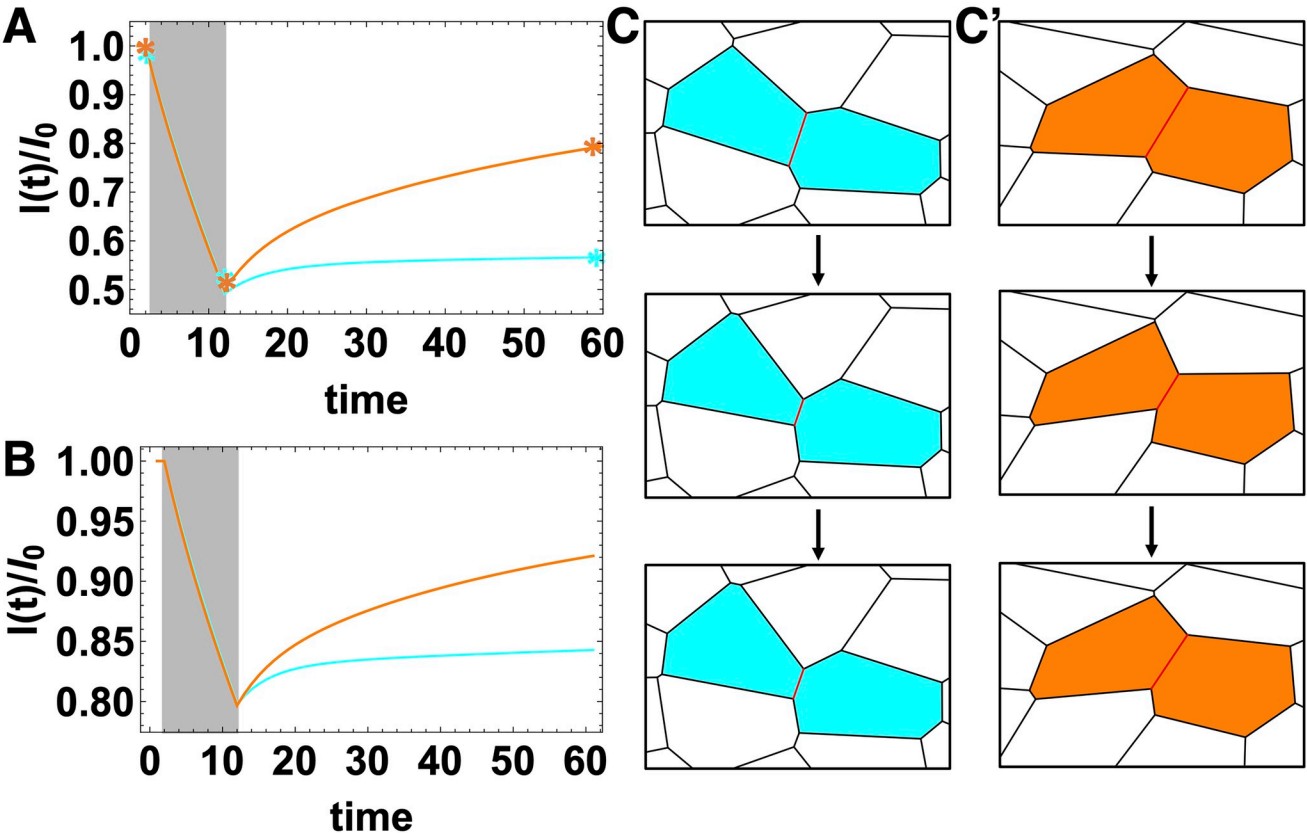

**Fig 6. Viscous response of the cellular junctions.** (A, B) Junction length, $l$, normalized by the initial length of the junction $l_0$ over time in units of simulation time steps, during and after a stress application on an edge in an anisotropic simulation. Grey regions indicate the time period of an high applied stress which shrinks the edge by %50 (A) and a low applied stress which shrinks the edge by %20 (B). The blue curves are from the simulations without a T1 delay, $t_{T1} = 0$ and the orange curves correspond to the simulations with a T1 delay time of $t_{T1} = 278.2\tau$. C) Snapshot from the simulations illustrating the edge dynamics before, during and after the high stress application (asterisks in (A) indicate the exact time points of the snapshots in (C) and (C')) with $t_{T1} = 0$ (C) or with $t_{T1} = 278.2\tau$ (C') T1 delay time. Other parameters are $\gamma_0 = 1.0$, $p_0 = 4.0$, $T = 0.0$ and $N = 256$.

orange curve) and low (Fig 6B orange curve) stress cases respectively. Even though the cell shapes are similar in both cases, the local viscous response of the cellular junctions change depending on the T1 delays in the tissue. Again, even at this smaller scale, systems with a larger T1 delay time are more elastic while those with a smaller T1 delay are more viscous.

## Discussion and conclusions

While standard vertex models for confluent tissues assume that T1 transitions proceed immediately after the configuration attains a multi-fold vertex, it is clear that some molecular processes may act as a brake on such transitions, generating a delay in the time required to resolve a higher-order vertex. In this work, we demonstrate that such T1 delays affect the tissue mechanical response in similar ways in isotropic, anisotropically sheared, and internally anisotropic tissues. Specifically, we demonstrate that the relaxation timescale associated with neighbor exchanges in the absence of T1 delays, $\tau_{\alpha 0}$, is an excellent metric for glassy tissue response in these disparate systems. Moreover, in systems with T1 delays, the observed relaxation timescale $\tau_\alpha$ is related to $\tau_{\alpha 0}$ in a remarkably simple manner. For $t_{T1} \lesssim \tau_{\alpha 0}$, the T1 delays do not strongly affect the system and $\tau_\alpha \sim \tau_{\alpha 0}$, while for $t_{T1} \gtrsim \tau_{\alpha 0}$ the T1 delay dominates the macroscopic dynamics and $\tau_\alpha = t_{T1}$. In a related observation, we find that the number of successful

T1 transitions, where cells neighbor exchange occurs and does not reverse at a later time, decreases significantly for $t_{T1} \gtrsim \tau_{z0}$.

This suggests that in tissues where molecular mechanisms generate large T1 delays, the standard vertex model picture—where tissue fluidity is correlated with cell shapes, adhesion, and cortical tension—breaks down. While such molecular mechanisms are not able to speed up rearrangements, they can generically slow them down, provided that the T1 delays are larger that the inherent relaxation timescale of the tissue. As an example, in processes such as convergent extension of the body axis in *Drosophila*, the aspect ratio changes by a factor of two in about 30 minutes [8]. Given that cells in a hexagonal vertex model would normally change neighbors after a change of about two in the aspect ratio [39, 40], our results suggest that in *Drosophila* germband extension, molecular processes that require on the order of tens of minutes or more to complete would be effective at interfering with global extension rates.

Interestingly, the behavior of T1 transitions over time for large T1 delays (Fig 5A) exhibits similar features to those observed in the germband of *Drosophila snail twist* and *bnt* mutant embryos [21, 41]. In such embryos the rearrangement rate does decrease significantly or disappear altogether, despite the fact that their cell shapes would suggest a high rearrangement rate in a standard vertex model [21]. This is consistent with the hypothesis that molecular mechanisms in these mutants act as a brake on T1 transitions across all of developmental time.

It is interesting to speculate that even in wild type embryos, such molecular brakes could be deployed at different stages of development to "freeze in" structures sculpted previously while the tissue was a fluid-like phase. For example, after the initial rapid elongation of the body axis in fruit fly described in the previous paragraph, the cellular rearrangement rates decrease fairly precipitously about 20 minutes after the elongation process initiates, even though the cell shapes are elongated and become even more so [21].

An additional observation is that in anisotropic systems, increased T1 delay times are associated with increased persistence of higher-fold coordinated vertices, which we track by identifying very short edges in our computer model. Specifically, for $t_{T1} \gtrsim \tau_{z0}$ we find that the number of very short edges per cell increases dramatically and remains high throughout the simulation. Again, this is consistent with the observation of a significant number of rosettes in the later stages of *Drosophila* body axes elongation [4].

In concurrent work, Das et al. [30] have studied a similar mechanism with an embargo on cell neighbor exchange time, for a constant target shape index parameter and in isotropic tissues. They discover interesting streaming glassy states where cells migrate in intermittent coherent streams, similar to what is seen in spheroid/ECM experiments. Our work is complementary, as we study both isotropic and anisotropic tissues over a range of shape index parameters. This allows us to emphasize the importance of the competition between the collective response timescale driven by cell-scale properties and the T1 delay timescale driven by molecular scale proerties at vertices. In addition, our focus on global anisotropic changes to tissue shape allows this work to serve as a starting point for understanding how T1 delays impact developmental processes such as the convergent extension and rosette formation during body axis elongation.

Although here we use the cell neighbor exchange timescale as a read-out for tissue fluidity, an interesting avenue for future work is studying how these observations correlate with explicit mechanical measurements. As the T1 delay time we introduce via the dynamics does not alter the mechanical energy of the vertex model, infinitesimal linear response observables calculated directly from the energy, such as the bulk and shear moduli, do not include any explicit contributions from the T1 delay. As a result, the zero-temperature static shear modulus is formally zero for many of the configurations we studied, even in cases where rearrangement rates are observed to be small. A more meaningful measure would be the nonlinear and dynamic

rheology of the tissue. A recent study examined the nonlinear rheology of the bare vertex model [42]; it would be interesting to extend these techniques to our system with T1 delay times. In colloids [43] it is well established that particle-motion-based measurements such as the mean squared displacement or the self-overlap function are directly related to the dynamic modulus measured in a rheometer, and it would be interesting to test this relationship in simulations and experiments on confluent tissue.

Taken together, our work suggests that moving forward it is really important to design experiments that investigate which types of molecular processes are acting as brakes on T1 transitions. Obvious candidates are players in the cooperative disassembly and reassembly of complex adhesive cell-cell junctions such as adherens junctions [44] and/or desmosomes [45], or dynamics of molecules such as tricellulin that localize to three-fold coordinated vertices [12].

In recent work, Finegan *et al*. [14] study *sdk* mutants that lack the adhesion molecule Side-kick(Sdk) which localizes at tricellular vertices. They show that *Drosophila sdk* mutants exhibit a 1-minute delay in cell rearrangement timescales compared to wt embryos during *Drosophila* axis extension, accompanied by more elongated cell shapes during the extension. In addition, they develop a vertex model that explicitly allows the formation of rosettes (higher fold-coordinated vertices) and additionally specifies that rosettes take longer to resolve than simple 4-fold coordinated vertices. The model recapitulates cell shapes and global tissue deformations seen in experiments. The spirit of the vertex model in that work is very similar to the one we report here, except that we do not require any special rules for rosettes. In our model they form naturally in systems where T1 delays occur, and they take longer to resolve simply because the individual vertices that comprise them each take longer to resolve. Therefore, it would be interesting to see if *sdk* mutants are quantitatively consistent with the model presented here, and whether one could estimate the T1 delay timescale by fitting to the model, and then look for molecular processes at the vertices that occur on that same timescale that might be driving the delay.

In related recent work, Yu and Zallen [46] study Canoe, a different tricellular junctional protein. They find that recruitment of Canoe to tricellular junctions is correlated with myosin localization during *Drosophila* convergent extension, and cells are arrested at four-fold vertex configurations in embryos that express vertex-trapped Canoe. The arrested cell rearrangements are 4 min longer compared to the rearrangements in wt embryos. In combination with our work, this suggests Canoe might also regulate T1 delays, and that tissues with perturbed Canoe dynamics or expression might another good system for testing our predictions about the relationship between T1 delays and global tissue mechanical properties.

Lastly, one particularly intriguing avenue suggested by recent work [16] is whether mechano-sensitive molecules may generate a stress-dependence for the T1 delay timescale. In other words, our work here focuses on the effects of a fixed T1 delay timescale that is independent of any local mechanical features of the cells. However, it is possible to introduce a feedback loop where T1 delays are longer or shorter depending on the magnitude of the stresses on nearby edges, mimicking the behavior of well-known catch or slip bonds except now at a cell- or tissue- scale. Such feedback loops could lead to interesting patterning and dynamical behavior.

## Supporting information

**S1 File. Supporting information file.**
(PDF)

## Acknowledgments

We thank Daniel Sussman, Matthias Merkel, Peter Morse, Karen Kasza, and Xun Wang for fruitful discussions.

## Author Contributions

**Conceptualization:** Gonca Erdemci-Tandogan, M. Lisa Manning.

**Data curation:** Gonca Erdemci-Tandogan.

**Formal analysis:** Gonca Erdemci-Tandogan, M. Lisa Manning.

**Funding acquisition:** M. Lisa Manning.

**Investigation:** Gonca Erdemci-Tandogan, M. Lisa Manning.

**Methodology:** M. Lisa Manning.

**Project administration:** M. Lisa Manning.

**Resources:** M. Lisa Manning.

**Software:** Gonca Erdemci-Tandogan.

**Supervision:** M. Lisa Manning.

**Validation:** Gonca Erdemci-Tandogan.

**Visualization:** Gonca Erdemci-Tandogan.

**Writing – original draft:** Gonca Erdemci-Tandogan, M. Lisa Manning.

**Writing – review & editing:** Gonca Erdemci-Tandogan, M. Lisa Manning.

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
