## [Decision Letter · Decision Letter 0]

24 Mar 2021

Dear Dr. Erdemci-Tandogan,

Thank you very much for submitting your manuscript "Effect of cellular rearrangement time delays on the rheology of vertex models for confluent tissues" for consideration at PLOS Computational Biology. As with all papers reviewed by the journal, your manuscript was reviewed by members of the editorial board and by several independent reviewers. The reviewers appreciated the attention to an important topic. Based on the reviews, we are likely to accept this manuscript for publication, providing that you modify the manuscript according to the review recommendations.

Sincerely,

Qing Nie

Associate Editor

PLOS Computational Biology

Jason Haugh

Deputy Editor

PLOS Computational Biology

[LINK]

Reviewer's Responses to Questions

**Comments to the Authors:**

Reviewer #1: This manuscript reports an implementation to the classical vertex model that will have an important impact for simulating remodelling of epithelial tissue morphogenesis. The authors include indeed a time delay that corresponds to the most sensitive (and still quite mysterious) step in cell intercalation, i.e. the famous T1 transition. This step corresponds to the phase where the shrinking contacts has brought the two vertices so close that they seemingly fuse into a single four-fold vertex that may either regress (aborted transition), or successfully resolve in neighbour exchange with formation of a new contact. So far, vertex models have considered this step as instantaneous. Yes, at the cellular level, this transition is certainly not a straightforward mechanism. One may rather expect it to be complex, time- and energy-consuming, thus .

The authors go on to explore the consequences of including this delay, and discover a very simple relationship between the length of this delay and the time scale of global tissue response. One very interesting consequence suggested by the authors is that by mechanisms that may slow down T1 transitions are predicted to stiffen the tissue. I would add that conversely, the still unknown molecular events required to remodel and/or dismount two tricellular junctions of an epithelium to build a new contact are bond to constitute a general rate limiting factor to epithelial morphogenesis.

While I can commend the high interest of this manuscript in the field of morphogenesis, I should specify that I am not a physicist, thus I am not able to critically evaluate the detailed mathematical aspects of the manuscript, although I do understand the general features of the model, the basic formulae and the implemented parameters, which I find fully adequate.

Reviewer #2: In their manuscript, Erdemci-Tandogan and Manning studied how cellular rearrangement delays might affect the dynamics of confluent tissues. In confluent tissues, where cells occupy the entire space, the neighbor exchange is the dominant mode for large-scale movement of cells in the tissue. In 2D, this process is modeled by so-called T1 transition, where the boundary between two cells shrinks to zero, the local connectivity of the four cells surrounding this boundary changes, which then follows by an expansion of the new boundary. In original Vertex models, this process is taken ad hoc, with no details that might reflect the rearrangement of proteins in a cell boundary going through a T1 transition. Such considerations are necessary to more precisely understand tissue dynamics. In their manuscript, Erdemci-Tandogan and Manning present a modification of the T1 transition, where the rearrangement is delayed. Such delay has several consequences on tissue remodeling, including the accumulation of four-way and higher coordination vertices, and more generally, the tissue dynamics. I found the results in this manuscript interesting and worth publishing in PLOS Computational Biology. However, I have a few suggestions and comments that might help the readers:

1- The authors used a time delay in T1 transitions by explicitly preventing the shrinking boundaries from going through rearrangement. Another way to include such delay would be to assign an energy barrier for the T1 transitions. This has the advantage of a slight modification to the original Vertex model energy function but keeping the dynamics unchanged. Could the authors elaborate if an explicit delay in T1 transitions is a more appropriate way of modeling the biological details of cellular rearrangements than other possible models such as an energy barrier?

2- I understand the origin of thermal noise in the context of molecular dynamic simulations of passive particles. However, I do not understand the origin of thermal fluctuations of vertices in the context of confluent tissues. Is this just a modeling concept, or is it referring to a biological process? Also, do the results fundamentally different at zero temperature? If not, why the authors did not consider studying their model in zero temperature, which is one less parameter to worry about.

3- In modeling anisotropic tension and tissue shear, the authors considered a fairly small box for such a large deformation. There might be finite-size effects due to periodic boundary conditions. I suggest the authors check this by repeating some of the simulations for different box sizes (i.e., number of cells) to ensure the results do not depend on box size.

4- While the authors performed a detailed analysis of T1 transitions in their simulations, it would be interesting to connect those to more coarse-grained properties of the network, such as shear and bulk moduli. Is it possible to measure the shear modulus of the network as a function of T1 delay, and if so, I recommend incorporating it in the manuscript.

5- In the simulations of anisotropic tension, the authors change the tension on a boundary based on its angle relative to an axis in the tissue. As the tissue evolves and the boundaries change orientation, their tension instantaneously changes. However, changes in the tension of a boundary require the actomyosin rearrangement, which also might happen with its timescales and delays

**Have all data underlying the figures and results presented in the manuscript been provided?**

Reviewer #1: Yes

Reviewer #2: Yes

PLOS authors have the option to publish the peer review history of their article (what does this mean?). If published, this will include your full peer review and any attached files.

Reviewer #1: No

Reviewer #2: No

Figure Files:

Data Requirements:

Reproducibility:

References:

---

## [Decision Letter · Decision Letter 1]

7 May 2021

Dear Dr. Erdemci-Tandogan,

We are pleased to inform you that your manuscript 'Effect of cellular rearrangement time delays on the rheology of vertex models for confluent tissues' has been provisionally accepted for publication in PLOS Computational Biology.

Best regards,

Qing Nie

Associate Editor

PLOS Computational Biology

Jason Haugh

Deputy Editor

PLOS Computational Biology

Reviewer's Responses to Questions

**Comments to the Authors:**

Reviewer #2: No further comments.

**Have the authors made all data and (if applicable) computational code underlying the findings in their manuscript fully available?**

Reviewer #2: Yes

PLOS authors have the option to publish the peer review history of their article (what does this mean?). If published, this will include your full peer review and any attached files.

Reviewer #2: No

---

## [Editor Report · Acceptance letter]

4 Jun 2021

PCOMPBIOL-D-21-00286R1 

Effect of cellular rearrangement time delays on the rheology of vertex models for confluent tissues

Dear Dr Erdemci-Tandogan,

I am pleased to inform you that your manuscript has been formally accepted for publication in PLOS Computational Biology. Your manuscript is now with our production department and you will be notified of the publication date in due course.

With kind regards,

Katalin Szabo
